# The Nuclear Pore Complex Is a Key Target of Viral Proteases to Promote Viral Replication

**DOI:** 10.3390/v13040706

**Published:** 2021-04-19

**Authors:** Luis Adrián De Jesús-González, Selvin Palacios-Rápalo, José Manuel Reyes-Ruiz, Juan Fidel Osuna-Ramos, Carlos Daniel Cordero-Rivera, Carlos Noé Farfan-Morales, Ana Lorena Gutiérrez-Escolano, Rosa María del Ángel

**Affiliations:** Department of Infectomics and Molecular Pathogenesis, Center for Research and Advanced Studies (CINVESTAV-IPN), Mexico City 07360, Mexico; luis.dejesus@cinvestav.mx (L.A.D.J.-G.); selvin.palacios@cinvestav.mx (S.P.-R.); jmreyesr@cinvestav.mx (J.M.R.-R.); jfosuma@cinvestav.mx (J.F.O.-R.); carlos.cordero@cinvestav.mx (C.D.C.-R.); carlos.farfan@cinvestav.mx (C.N.F.-M.)

**Keywords:** viral proteases, nuclear pore complex, flavivirus, enterovirus, nucleus

## Abstract

Various viruses alter nuclear pore complex (NPC) integrity to access the nuclear content favoring their replication. Alteration of the nuclear pore complex has been observed not only in viruses that replicate in the nucleus but also in viruses with a cytoplasmic replicative cycle. In this last case, the alteration of the NPC can reduce the transport of transcription factors involved in the immune response or mRNA maturation, or inhibit the transport of mRNA from the nucleus to the cytoplasm, favoring the translation of viral mRNAs or allowing access to nuclear factors necessary for viral replication. In most cases, the alteration of the NPC is mediated by viral proteins, being the viral proteases, one of the most critical groups of viral proteins that regulate these nucleus–cytoplasmic transport changes. This review focuses on the description and discussion of the role of viral proteases in the modification of nucleus–cytoplasmic transport in viruses with cytoplasmic replicative cycles and its repercussions in viral replication.

## 1. Introduction. The NPC as a Target for Viral Proteases: Controlling the Nucleus–Cytoplasmic Transport

Most viruses with nuclear and cytoplasmic replication require nuclear proteins to facilitate their replication [1]. Some viral proteins from viruses with cytoplasmatic replication localize in nuclei during infection. Moreover, the inhibition of nuclear transport of viral proteins using different drugs targeting the transport machinery has been shown to inhibit viral replication, suggesting that viral proteins have essential roles in the nucleus of the infected cells [2].

Different positive-strand RNA viruses modify the nucleus–cytoplasmic traffic, altering the Nuclear Pore Complex (NPC) through various mechanisms (Table 1). These alterations are carried out to (a) access nuclear proteins that participate in viral replication in the cytoplasm, (b) modulate the antiviral immune response by preventing the trafficking of transcription factors to the nucleus, and (c) negatively regulate the export of cellular mRNA to the cytoplasm by avoiding competition for the translation machinery and reducing the translation of proteins related with the antiviral response [2,3,4,5,6,7,8].

Since it has been well established that disruption of the import–export mechanism is a checkpoint for virus replication, it is not surprising to find that viruses induce the degradation of some components of the NPC, such as nucleoporins (Nups), by the action of their own viral proteases. This review discusses how proteases of positive-strand RNA viruses from the genera *Enterovirus*, especially *Poliovirus* and *Rhinovirus* (Family *Picornaviridae*), and *Flavivirus* (Family *Flaviviridae*), affect the composition of the NPC and the consequences for nucleus–cytoplasmic traffic through the processing of Nups.

## 2. NPC Structure and Function in the Cell: One Way to Access the Nucleus

### 2.1. The Machinery of The Nuclear Pore Complex (NPC)

The import and export of viral proteins requires interaction with the components of the nuclear envelope (NE), which is formed by a double membrane that harbor protein channels called nuclear pore complexes (NPCs). NPCs are formed by multiple copies of Nups that participate in the bi-directional nucleus–cytoplasmic transport of macromolecules, ribosomal subunits, viral proteins and RNAs (mRNAs, rRNAs, tRNAs, miRNAs) from both cellular and viral origin [29,30,31].

The NPC is made up of approximately 30 multi-copy of Nups, arranged from the cytoplasm to the nucleoplasm, which makes up the cytoplasmic filaments, cytoplasmic ring, internal pore ring, nuclear ring, and nuclear basket [32,33]. Inside the NPC, there are Nups with repeating sequences rich in phenylalanine (Phe) and glycine (Gly), called FG-Nups, such as Nup358, Nup62, Nup58, Nup54, Nup98, Nup45, Nup214, hCG1, Nup153, TPR, and Nup50, that facilitate the nuclear transport in both directions since they join to nuclear transport receivers (NTRs) (Figure 1) [34,35,36].

### 2.2. Bidirectional Nucleus–Cytoplasm Transport

Molecules smaller than approximately 40–50 kDa can pass freely through the nuclear envelope; however, higher molecular weight molecules such as proteins and RNAs from both cellular and viral origin are actively transported through the NPC, between the nucleoplasm and the cytoplasm. The nuclear import and export of molecules are regulated by NTR (nuclear transport receptors), such as importins, exportins, carriers, and small GTPases of the Ran family that regulate the activity of importins and exportins that transport cargo molecules (Figure 2) [36,37,38].

NTRs recognize specific sequences in cargo proteins that cross the nuclear membrane from the cytoplasm, such as nuclear location sequences (NLS) that contain repeated arginine (Arg or R) and lysine (Lys or K) amino acids. The classical NLS consists of five KKKRK amino acids. Moreover, some proteins possess bipartite NLS consisting of two groups of basic amino acids, separated by approximately ten amino acids [39,40]. On the other hand, nuclear export sequences (NES) participate in the trafficking from the nucleus to the cytoplasm. They are composed of sequences rich in leucine or hydrophobic amino acids such as valine (Val), isoleucine (Ile), phenylalanine (Phe), or methionine (Met), which are found in motifs conserved in cargo proteins, such as some transcription or translation factors and mRNA transport proteins [41,42,43]. The nuclear localization of a given protein occurs by the recognition of the NLS by the NTRs; for example, importin α through its NTR binds to NLS-cargo, then importin β binds to importin α to form a trimeric complex and its cargo molecule. If the NLS is atypical, then importin β directly binds to its cargo molecule without the participation of importin α. The directionality of the cargo is given by the Ran-GTP gradient, regulated by the Ran-GTP/GDP cycle. Once the trimeric complex has entered the nucleus, the Ran-GTP activated by RCC1 (GEF) joins importin β and thereby is released from its cargo. Importin β is transported to the cytoplasm, and the Ran-GTP is deactivated to Ran-GDP by GAP (GTPase Activating Protein) to free itself from importin β for its next import cycle.

On the other hand, the nuclear export is given by the recognition of the nuclear export sequences (NES). Nuclear export begins with Ran-GTP binding to exportin (e.g., CRM-1), which causes an increased affinity for the export cargo. Then, the complex moves to the nuclear pore and Ran-GTP hydrolyze (activated by RCC1), which forms the export complex. The complex crosses the NPC, and in the cytoplasm, GAP deactivates RanGTP (hydrolyses the GTP in GDP), causing the export protein to be released from its cargo [39,40,44].

## 3. *Flavivirus*: A Genus of Medical Importance

*Flavivirus* is a genus belonging to the family *Flaviviridae*, which contain viruses such as SPONV (Spondweni virus), YFV (Yellow fever virus), WNV (West Nile virus), JEV (Japanese encephalitis virus), DENV (Dengue virus), and ZIKV (Zika virus), widely distributed in tropical and subtropical areas. [44,45,46,47,48,49]. These viruses, transmitted by mosquitoes of the genus *Aedes* spp. [44], causes diseases with high morbidity and mortality; therefore, of medical importance.

Flaviviruses contain RNA genomes of positive sense that encode a polyprotein that gives rise to three structural proteins (C, E, and prM) present in the viral particles and seven non-structural proteins (NS1, NS2A, NS2B, NS3, NS4A, NS4B, and NS5) essential for controlling several cell pathways for the replication of the viral genomes, assembly, and release of the viral particles from the infected cells (Figure 3A) [50,51,52,53].

*Flavivirus* has a cytoplasmic replicative cycle, translating and replicating its genome in the ER [54,55,56]. However, some viral proteins, such as C, NS1, and NS5, contain nuclear location signals (NLS) and have been seen located in the nucleus of Vero cells [57,58,59]. Other proteins located in the nucleus of human lung carcinoma (A549) infected cells are NS2A, NS3, and NS4A; moreover, the NS3 protein also locates in the nucleus of Huh7 and C636 cells during DENV infection. [60,61,62,63].

The NS3 protein is the only protease encoded by these viruses and is required for the cleavage of the viral polyprotein and the unwinding of the viral RNA since it has the function of protease and helicase. Since this molecule is involved in the infection establishment, it has been used as a pharmacological target against the different *Flavivirus* infections [64,65,66].

### The Non-Structural Protein 3 (NS3): A Shuttling Protein

The NS3 protein of *Flavivirus* is a trypsin-like serine protease. It requires a 40-residue hydrophilic segment of the NS2B transmembrane protein as a cofactor, and it is involved in the processing of the polyprotein that gives rise to the mature viral proteins [67,68].

NS3 protease has a catalytic triad conserved in the different members of the genus (His51, Asp75, and Ser135) and a total length of approximately 620 amino acids. NS3 protease cleaves between a basic residue (Arg-Lys) and another amino acid such as serine or threonine. In addition to the protease function, at its carboxyl terminus, NS3 has RNA triphosphatase, nucleoside triphosphatase, and helicase activities (Figure 3B). The conservation of the amino acid sequence from NS3 among the different Flaviviruses (DENV, WNV, JEV, ZIKV, and YFV) is between 50% to 75% [68,69,70,71].

NS3 from ZIKV is mainly located in the perinuclear region of the infected cell, and alters the morphology of the nuclear lamina, a component of the nuclear envelope, forming extrusion sites; thus, it is suggested that it may affect the function of the centrosome [58]. NS3 is also accumulated in the concave face of the kidney-shaped altered nuclei and may be responsible for modifying other components of the nuclear envelope [60]. Our research group has recently found that NS3 of DENV is located in the nucleus of DENV infected Huh7 and C636 cells at early times post-infection (8 to 12 h) and in the cytoplasm at later times (16 to 24 h post-infection) [61,62].

In addition to the polyprotein processing, NS3pro from several Flaviviruses have other cellular proteolytic targets. Hill et al. have reported that NS2B-3 from ZIKV can process 31 different cellular proteins, such as JIP4, ATG16L1, eIF4G1, TAK1. Among them, 42% are involved in the processing of genetic information, 28% in transport and the cytoskeleton, 12% in metabolism, 10% in signal transduction, and 8% in immunity [69]. On the other hand, NS3pro from ZIKV and DENV also cleaves FAM134B (related to reticulophagy) [72]; ZIKV NS3pro cleaves Septin-2 (cytoskeletal factor, related to cell cytokinesis) [73], PDIA3, ALDOA (glycolysis) [70]; and NS3pro cleaves DDX21 (immune response) (DENV) [71] and GrpEL1, (mitochondrial matrix protein) [74].

*Flavivirus* NS3pro also cleaves components of the NPC [12], a subject of this review; these alterations occur to guarantee its replication in the host cell.

## 4. Alterations of the NPC by NS2B-3 Protease of *Flavivirus*: The Cleavage of the Complex

We have recently described the alteration of the NPC during the infection with ZIKV, DENV-2, and DENV-4. We found that NS2B-3 from DENV-2 and DENV-4 cleaves Nup153 (located in nuclear basket), Nup98 (located in cytoplasmic filaments and internal pore ring), and Nup62 (located in the inner pore ring). On the other hand, we found that NS2B-3 from ZIKV cleaves TPR (located in nuclear basket), Nup153, and Nup98 (Figure 3C). Besides, we found that the subcellular location of some Nups was also affected. The role of the cleavage/degradation of the NPC during infection with *Flavivirus* is still unknown [12]. However, during infection with other RNA-viruses such as Poliovirus and Rhinovirus, the degradation of nucleoporins has been associated with the inhibition of the import/export of transcription factors, mRNAs, and immune response modulation to guarantee viral replication [75].

### Nucleus Involvement during Flavivirus Infection: A Brief Overview

*Flavivirus* infections cause a spectrum of diseases, including fever, hepatitis, vascular shock syndrome, neurological and congenital abnormalities, and encephalitis [76].

Although the Flaviviruses have a cytoplasmic replication cycle, it has been observed that some DENV and ZIKV proteins such as C, NS1, and NS5 are located in the nucleus of Vero cells. The transport to the nucleus of the viral proteins is mediated by the NLS present in its sequence [57,58,59]. In the same way, the C, NS3, NS4B, and NS5 proteins of other *Flavivirus* such as JEV, WNV, and DENV have also been observed in the nucleus of infected cells [62]. However, little is known about the role of these proteins in the nucleus. Interestingly, mutations in the NLS of C and NS5 reduce the production of virions suggesting that its presence in the nucleus is essential during the viral replicative cycle [2,77,78,79,80,81].

Besides internalization of viral proteins into the nucleus during *Flavivirus* infections, some nuclear components are highjacked to avoid the antiviral immune response. The presence of the NS5 polymerase of WNV, DENV, and JEV in the nucleus mediates the IFN response pathway at different levels of the JAK/STAT signaling pathway and thereby prevents the entry of transcription factors into the nucleus to avoid the innate immune response [82,83,84,85].

On the other hand, nuclear proteins such as La (DENV), PTB (DENV), RNA Helicase A, TIA1/TIAR (WNV), and Tudor-DN/p100 (DENV) migrate to the cytoplasm to participate in the assembly of the viral replication machinery [86,87,88,89,90,91].

Despite the knowledge of the presence of nuclear components in the viral replicative complexes, the mechanisms involved in this process are still unknown [2].

## 5. *Picornaviridae*: A Family of Small Viruses

The *Picornaviridae* family is formed of at least 30 genera, including the *Enterovirus* genus, and more than 75 species [92].

Viruses of the *Picornaviridae* family have a single-stranded RNA genome of positive polarity of 6.7 to 10.1 kb in length. They are small, non-enveloped viruses, which infect several vertebrates, such as mammals, birds, reptiles, amphibians, and fish. They are also human pathogens of medical relevance because they affect the central nervous system, heart, liver, skin, gastrointestinal tract, and upper respiratory tract [92,93,94].

The replicative cycle of the *Picornaviridae* is cytoplasmic, and the RNA genome is translated and replicated in the endoplasmic reticulum (ER). The genome of the *Picornaviridae* is composed of a single open reading frame (ORF), which codes for a polyprotein that, when cleaved by viral proteases, gives rise to three to four structural proteins that conform the capsid and seven non-structural proteins involved in viral replication and the modulation of the immune response. The genome at its 5′ end is covalently linked to a VPg protein involved in RNA replication and is polyadenylated at its 3′ end. Two untranslated regions (UTR) containing RNA stem-loop structures involved in the regulation of replication flanked the unique ORF; particularly, the 5′ UTR, of 1/10 of the total genome length promotes viral protein synthesis through an internal ribosome entry site (IRES). This polyprotein is processed by the viral proteases to produce P1–P3 regions that are further processed to produce precursor and mature proteins. The P1 region conformed by the structural proteins (VP1-VP4), and the P2 (2A–2C) and P3 (3A–3D) regions give rise to non-structural proteins. Intermediate protein precursors are also produced, having essential roles, such as the 3CD protease-polymerase protein (Figure 4A) [92,95,96,97,98,99].

Both precursor and mature proteins produced by the proteases cleavage participate in viral replication, virion structural formation and modify the host immune response [100,101].

### 5.1. Enterovirus

The best-studied group of viruses altering the NPC by viral proteases belongs to the *Enteroviruses* genus, which comprised at least 300 different types of viruses, which cause diseases in humans (Rhinovirus A, B, C, and Enterovirus A–D) [101,102].

Although enteroviruses are cytopathic and mainly infect the gastrointestinal or the respiratory tracts, they can also infect the central nervous system like Poliovirus (PV). PV infects the gastrointestinal tract, where the infection is asymptomatic; however, it also has a tropism for motor neurons of the lower extremities, generating paralytic poliomyelitis in approximately 1% of the infected individuals [103,104]. On the other hand, Human Rhinoviruses (HRV) is the leading cause of almost half of the common human cold. They infect the upper respiratory tract and, in some cases, can cause chronic lung diseases, asthma, severe bronchiolitis in infants, and fatal pneumonia in adults. HRVs are classified into HRV-A, HRV-B, and HRV-C [105,106].

Particularly in the next section, we will discuss the alterations of the NPC by the viral proteases during the infection of the *Enterovirus* genus members, specifically PV and HRV.

### 5.2. Viral Proteases of Poliovirus and Rhinovirus

Unlike Flaviviruses, the polyprotein of both PV) and RV) is cleaved into functional proteins by two viral proteases: 2Apro and 3Cpro. While 2Apro cleaves P1 from P2 and P3, 3C pro is responsible for the cleavage of almost all precursors and mature proteins. Moreover, besides the polyprotein cleavage, both proteases are involved in establishing the infection [96].

#### 5.2.1. Apro

2Apro from PV and RV participates in the polyprotein processing at the P1 (VP1-2A) site (Figure 4B). Initially, 2Apro splits itself at its N-terminal site, and 3Cpro is involved in the cleavage of 2Apro at its C-terminal. 2Apro is a chymotrypsin-like cysteine protease, with catalytic triad His18, Asp34, and Cys105. Its structure comprises four antiparallel β-sheets in the N-terminal region and a β-barrel in the C-terminal region, and a length of 140–150 aa [107,108].

In addition to the polyprotein cleavage, 2Apro has other cleavage targets, such as the translation initiation factors (eIF4G-I and eIF4G-II), necessary to recognize the cap in the mRNAs by the ribosomes [109]. Therefore, by cleaving these initiation factors, enteroviruses disrupt cap-dependent translation and highjack the functional ribosomes to initiate viral synthesis via an independent cap mechanism known as IRES-dependent translation. On the other hand, in PV-infected HeLa cells, 2Apro cleaves cytokeratin 8, favoring the cytopathic effect and cell lysis to release the viral particles [110]. In the case of Coxsackievirus B3, PV, and Enterovirus 71, 2Apro has proteolytic activity on MDA5 (melanoma differentiation-associated protein 5) and MAVS (Mitochondrial antiviral-signaling protein); this action leads upstream to the blockage of IFN-I transcription and interferes with the innate immune response [111]. Also, 2Apro from PV and HRV participate from its perinuclear subcellular localization in altering the nuclear–cytoplasmic traffic through the processing of Nups, components of the NPC [3,7]. NPC disruption results in the presence of nuclear proteins in the cytoplasm of the infected cells, such as La and the PTB, considered among others, as internal translation associated factors (ITAFs) crucial for the regulation of IRES dependent translation.

#### 5.2.2. Cpro

As 2Apro, 3Cpro is a chymotrypsin-like cysteine protease; however, it adopts a fold-like conformation of the serine proteases, responsible for the processing of different proteolytic targets [96]. The structure of 3Cpro of PV contains two β barrel domains (6 antiparallel β sheets), opposite each other, and a length of 183 residues. The binding to the substrate occurs between both domains and contains the catalytic triad His40, Glu71, and Cys147. The polypeptide loop that precedes Cys147 is flexible and undergoes a conformational change after binding to the substrate [112]. 3Cpro cleaves at Gln/Gly residues, and it is responsible for most of the cuts in the polyprotein within P1 to produce VP0, VP1 and VP3, P2 (C-terminal 2A, 2B and 2C), and P3 (3A, 3B and 3C) sites (Figure 4C) [113].

Besides its role in the polyprotein precursor cleavage, 3Cpro has different proteolytic targets that affect many cell pathways at various levels to promote its viral replication and the establishment of the infection. 3Cpro can locate in the nuclei of the PV infected cells and inhibits host RNA synthesis by processing cell transcription factors, such as factor 110 associated with TATA-binding protein (TAF110), TATA box-binding protein (TBP), binding protein to the response element to cAMP-1 (CREB-1), octamer-binding protein-1 (Oct-1), p53 and transcription factor IIIC (TFIIIC) [114,115,116,117,118,119]. It causes a reduction of mature RNA molecules released into the cytoplasm, and therefore a decrease in molecules competing with the viral RNAs to be translated. 3Cpro from PV cleaves the poly-A binding protein (PABP) and the nuclear proteins PTB and La; in the latter, 3Cpro separates the NLS of the protein, allowing its cytoplasmic localization and participation in IRES translation [120,121]. Moreover, 3Cpro also processes cytoplasmic proteins such as translation initiation factor 5B (eIF5B) and the ITAF poly (rC) binding protein 2 (PCBP2), that, when processed, loses its translation function but maintains its role in RNA replication [122,123,124,125].

The cleavage of RIG-I by 3Cpro from PV and RV can block the recognition of viral RNA by the host’s innate immune system [111,126]. On the other hand, 3Cpro from PV can also affect the integrity of the cellular cytoskeleton through the cleavage of microtubule-associated protein 4 (MAP-4) [127].

Furthermore, 3Cpro from PV and RV also alters the nucleus–cytoplasmic traffic by processing nucleoporins, such as Nup62, Nup153, Nup214, and Nup358, components of NPC [4,16,17,19].

As mentioned above, the viruses use these strategies to favor their replication by preventing the activation of genes related to the antiviral immune response and avoiding competition with cellular mRNA by the translational machinery.

## 6. Alterations of the NPC by 2Apro y 3Cpro of *Poliovirus* and *Rhinovirus*: The Cleavage of the Complex

NPC alterations during viral infections due to specific cleavage of Nups by cysteine proteases with a chymotrypsin-like activity of cytoplasmic viruses such as PV and RV have been reported. The nucleoporins that are most commonly targets of these proteases are Nup358, Nup214, Nup153, Nup98, and Nup62, which results in the alteration of the nucleus–cytoplasmic transport of proteins and mRNAs [3,4,8,100], the reduction of cellular mRNA translation, seizing the translation machinery for viral replication and antiviral immune response control (Figure 5) [5,6,7,8,17,128,129].

### 6.1. Inhibition of the Nucleus–Cytoplasmic Trafficking of Proteins and RNA during PV Infection

Some authors have reported that during infection with PV, the nuclear proteins La, Sam68, and nucleolin were delocalized towards the cytoplasm [130,131,132]. Given this, in 2001, Gustin and Sarnow [4] analyzed the effects of infection with PV on the nucleus–cytoplasm traffic.

To analyze the classical nuclear import pathway (importin α/β), these authors used chimeric EGFP proteins coupled with the SV40 large T antigen NLS. Using confocal microscopy, they found that during infection with PV, the EGFP-NLS signal was located mainly in the cytoplasm, contrary to control uninfected conditions where the signal was in the cell nucleus. On the other hand, they analyzed the non-classical transport pathways, such as the one mediated by transportin 1 during infection. Transportin 1 participates in the bidirectional transport of proteins containing sequence M9-NLS (also called PY-NLS), present in proteins such as the heterogeneous nuclear ribonucleoprotein A1 (hnRNP A1), and some others involved in RNA binding, transcription, or processing [133]. When this pathway was analyzed during PV infection, Gustin and Sarnow found that EGFP-NLS-M9 was also redistributed from the nucleus to the cytoplasm in infected cells.

Furthermore, a battery of antibodies directed against proteins that use different transport routes was used to analyze their location. The hnRNP A1 (transportin 1 pathway [133]), hnRNP K (Importin α/β pathway and K nuclear-shuttling (KNS signal) [134]), and hnRNP C (nuclear retention sequence (NRS) was relocated to the cytoplasm during PV infection [135]). However, some pathways were not affected by infection, such as the Exportin CRM1 pathway and the Transportin SR pathway (demonstrated by the cellular location of SC35 [136]). These results show that some transport pathways are affected, and others remain functional during PV infection.

A necessary mediator for nucleus–cytoplasmic transport is the NPC, which was analyzed during infection with PV. Using the monoclonal antibody 414 (directed against FG-Nups) and specific antibodies against Nup153 and Nup62, a dramatic reduction in these proteins was observed in infected cells [4].

In 2008, this same group described that 2Apro from PV is responsible for Nup153, Nup98, and Nup62 cleavage; therefore, for the nuclear–cytoplasmic traffic during infection with PV [6]. Later in 2009, Castelló et al. reported that the cleaving of Nup153, Nup98, and Nup62, by 2Apro, interferes with the traffic of mRNAs, rRNAs, and U snRNAs from the nucleus to the cytoplasm, without any apparent effect on tRNA transport (Figure 5).

### 6.2. Inhibition of Nuclear–cytoplasmic Trafficking of Proteins and RNA during Infection with RV

Like PV, RV requires host nuclear proteins such as nucleolin, La, and Sam68 for its replication [5]. Given this fact, Gustin and Sarnow analyzed whether RV could alter the NPC integrity and its effect on nucleus–cytoplasmic transport.

As in PV infection, during the RV infection, Nup153 and Nup62 were also degraded, and consequently, the nuclear–cytoplasmic traffic was altered. Importin α/β (classic NLS), transportin 1 (M9-NLS), hnRNP A1 (transportin 1 pathway [133]), hnRNP K (Importin α/β pathway and K nuclear-shuttling (KNS signal) [134]), and hnRNP C (nuclear retention sequence (NRS) pathways were altered [135]), while others (such as Transportin SR pathway [136]) were not affected.

Furthermore, it was found that 2Apro of HRV-2 is responsible for cleaving Nup62 and thus altering the nuclear–cytoplasmic traffic [8]. In addition to these findings, Watters and Palmenberg reported that 2Apro from various RVs (HRV-A, HRV-B, and HRV-C) cleaves Nup153, Nup98, and Nup62 at different sites [7].

HRV 3Cpro and its precursor 3CD also participate in the cleavage of Nups and the alteration of the nuclear–cytoplasmic traffic. 3Cpro and 3CD degrade Nup153, Nup214, and Nup358 [16]. Furthermore, 3Cpro also downgrades Nup62 and Nup98 [17]

FG-Nups (Phe/Gly repeats) are recognized by the nuclear–cytoplasmic transport machinery (carioferins) [29]. Thus, the cleavage of the Nups by the proteases 2Apro and 3Cpro inhibits the transport activity. The proteolytic activity of 2Apro on NPC inhibits the importin α/β, transportin 1, transportin 3 (SR), and export pathways Crm1 (Figure 5) [137].

## 7. Conclusions

The consequences of the inhibition of transport can occur at the antiviral level through the inhibition of transport via importin α/β of NF-KB [138], IRF-3 [139], and STAT-1 [140]; at the translational level, through changes in the subcellular localization of nuclear proteins to replication centers, such as La, PTB, nucleolin, and Sam68; in RNA biosynthesis, by inhibiting the nuclear import of hnRNPs and SR proteins and by avoiding competition for the translational machinery by hampering the export of mRNAs.

The viral proteases of RNA-viruses are indispensable proteins for viral progeny production since they are the major components for the processing and maturation of viral proteins. Moreover, viral proteases also have essential roles in viral infection; for that, proteases alter the nucleus–cytoplasmic transport by targeting Nups in the NPC; thus, modifying different cellular pathways to favor viral replication and to evade the immune response. Thus, viral proteases constitute an attractive pharmacological target for the design of antiviral drugs.

In this review and other related works, it becomes clear that there is a need to not only study the participation of the cell nucleus and its components but also the involvement of viral proteins in the nucleus during the promotion of viral replication and infection of RNA-viruses and, thus, to influence in the development of drug targets. Viral proteases also have essential roles in viral infection.

## Figures and Tables

**Figure 1 viruses-13-00706-f001:**
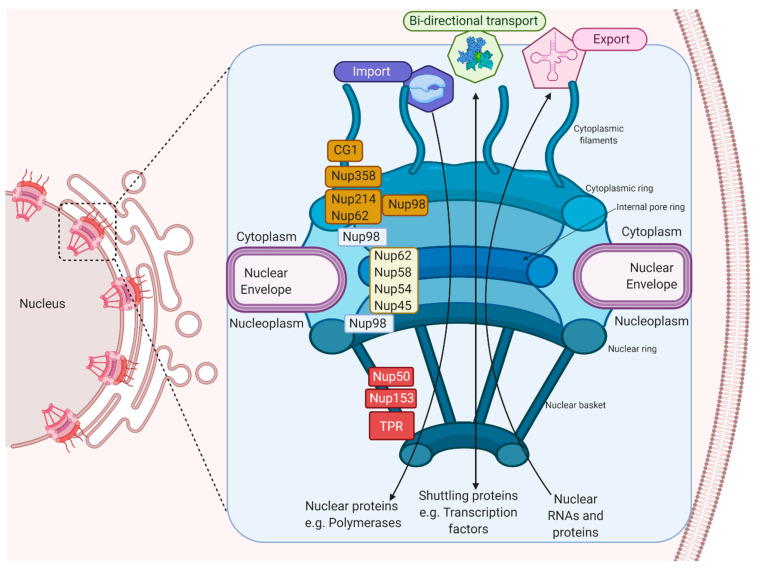
The machinery of the Nuclear Pore Complex. The NPC’s different components, which participate in the import and export of proteins and RNAs, and the FG-Nups, which participate in nuclear–cytoplasmic transport, are shown.

**Figure 2 viruses-13-00706-f002:**
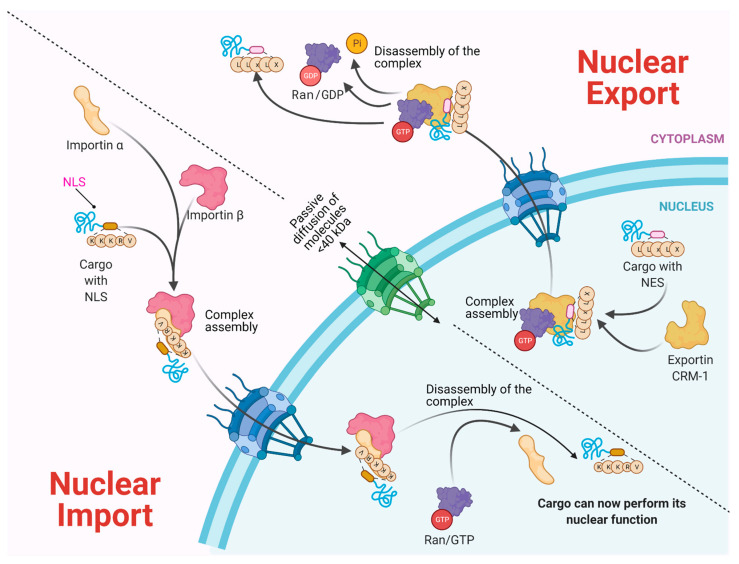
Bidirectional Nucleus–cytoplasm Transport. In the classical pathway of import (importin α/β), the importins and cargo complex are formed through the recognition of the NLS (nuclear location sequence). In the case of Export via CRM-1, the charge contains an NES (nuclear export sequence). In both cases, a GTP gradient is required. Molecules <40 kDa pass through passive diffusion towards the nucleus by the NPC.

**Figure 3 viruses-13-00706-f003:**
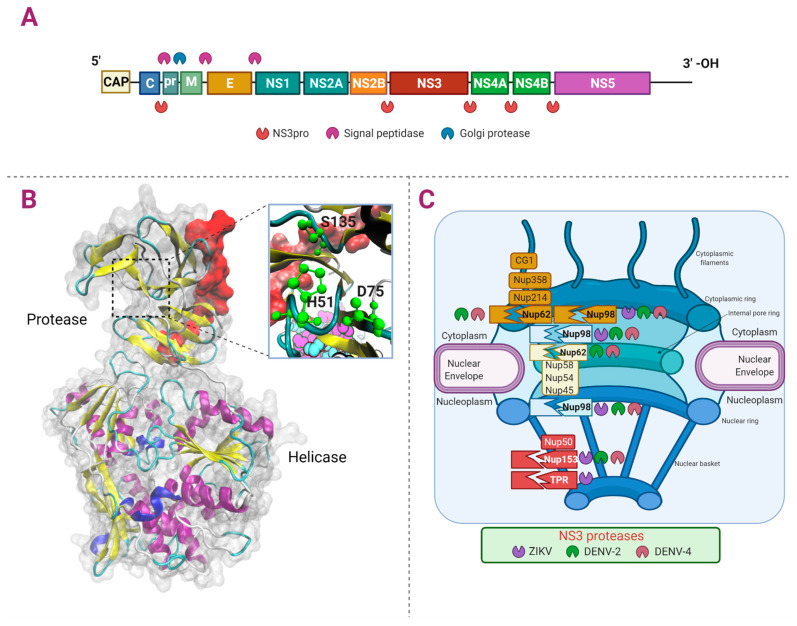
Proteolytic targets of *Flavivirus* NS3. Panel (**A**) shows the *Flavivirus* RNA genomic organization encoding three structural and seven non-structural proteins. Panel (**B**) shows the NS3 from DENV-4 (Protease and Helicase in New Cartoon (helix-α in purple, 3_10_ helix in blue, β-sheets in yellow, β-bridge in tan, and turn in cyan), and the zoom of the catalytic triad H51, D75, and S135 in green)). Also, cofactor NS2B (red) is shown (PDB ID: 2VBC). Panel (**C**) shows the cleavages of Nups by NS3pro from DENV and ZIKV.

**Figure 4 viruses-13-00706-f004:**
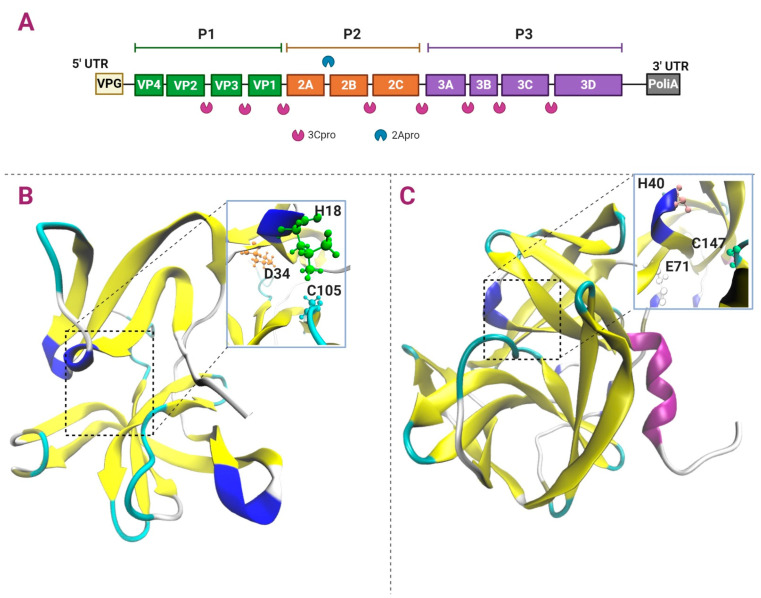
Polyprotein cleavage and structure of *Enterovirus* 2Apro and 3Cpro. Panel (**A**) shows the *Enterovirus* RNA genome that encodes four structural and seven non-structural proteins and the cleavage sites by 2Apro and 3Cpro. Panel (**B**) shows 2Apro and its catalytic triad H18, D34, and C105, from RV (PDB ID: 2M5T). Panel (**C**) shows 3Cpro and its catalytic triad H40, E71, and C147, from PV (PDB ID: 1L1N). Color code: helix-α in purple, 3_10_ helix in blue, β-sheets in yellow, β-bridge in tan, and turn in cyan.

**Figure 5 viruses-13-00706-f005:**
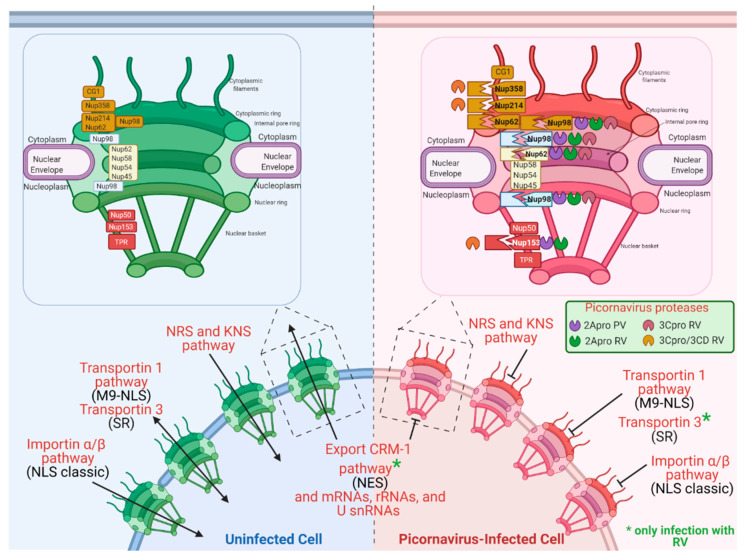
NPC alterations by Poliovirus and Rhinovirus 2Apro and 3Cpro activity and cleavage consequences. Nups cleavages made up by 2Apro, and 3Cpro of RV and PV are shown. Some nuclear import and export pathways are shown in uninfected and RV and PV-infected cells. Inhibition of pathways of import and export nuclear are shown.

**Table 1 viruses-13-00706-t001:** Mechanisms of positive-strand RNA viruses to affect the nucleus–cytoplasmic trafficking.

Group Baltimore	RNA-Virus (Family)	Viral Protein	NPC Alteration Mechanism
IV (ssRNA+)	SARS-CoV (*Coronaviridae*)	NSP1	Nup93 delocalization to nucleoplasm [9].
	SARS-CoV-2 (*Coronaviridae*)	ORF6	Delocalization of RAE-1 and Nup98 to the cytoplasm and nuclear accumulation of hnRNP A1 [10]
	SARS-CoV-2 (*Coronaviridae*)	ORF6	Hijacking the RAE-1-Nup98 complex to prevent the nuclear translocation of STAT-1 and 2 and the IFN response [11].
	DENV (*Flaviviridae*)	NS3pro	Cleavage of Nup153, Nup98, and Nup62 by the viral protease [12].
	ZIKV (*Flaviviridae*)	NS3pro	Cleavage of TPR, Nup153, and Nup98 by the viral protease [12].
	Porcine reproductive and respiratory syndrome virus (*Arteriviridae*)	Nsp1β	Alteration of the immune response through the interaction with Nup62 interrupts the import of transcription factors related to IFN [13].
	Encephalomyocarditis virus (*Picornaviridae*)	Leader	Inhibition of the nucleus–cytoplasmic trafficking by hyper-phosphorylation of Nup62, Nup153, and Nup214 [14].
	Theiler’s murine encephalomyelitis virus (*Picornaviridae*)	Leader	Blocking the cellular mRNA export and promotion phosphorylation of Nup98 [15].
	Rhinovirus (*Picornaviridae*)	2Apro	Cleavage of Nup62, Nup98, Nup153, Nup98 by the viral protease alters nucleus–cytoplasmic transport [3,5,8,16,17].
	Rhinovirus (*Picornaviridae*)	3C pro/3CD	Cleavage of Nup358 (TPR), Nup214, Nup153, and Nup62 by viral protease alters nucleus–cytoplasmic transport [16,17].
	Poliovirus (*Picornaviridae*)	2Apro	Cleavage of Nup153, Nup98, Nup62 by viral protease alters nucleus–cytoplasmic transport [4,6].
V (ssRNA−)	Rift Valley fever virus (*Phenuiviridae*)	Non-structural protein	Alteration of the antiviral response due to the degradation of Nup62. Besides, the participation of Nup98 in the nuclear import of NSs and in viral replication [18].
	Influenza A virus (*Orthomyxoviridae*)	NS2	Interaction with Nup214 for the export of viral RNA [19].
	Influenza A virus (*Orthomyxoviridae*)	NS2	Interaction with Nup98 to promote viral spread [20].
	Influenza A virus (*Orthomyxoviridae*)	-	Regulation of the antiviral immune response by lowering Nup98 and Rae1. Deterioration of mRNA export [21].
VI (ssRNA-RT)	VIH-1 (*Retroviridae*)	-	Relocation of hnRNP A1 and vRNA to the cytoplasm by the negative regulation of Nup62 [22].
	VIH-1 (*Retroviridae*)	Capsid	Participation of RanBP2/Nup358, Nup153, and Nup98 in anchoring the capsid to the nucleus [23,24,25,26,27,28].

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
