# Peer review of "The Nuclear Pore Complex Is a Key Target of Viral Proteases to Promote Viral Replication"

_viruses, 2021, doi:10.3390/v13040706_

Round 1

Reviewer 1 Report

In this review manuscript, Jesús-González et al. aim at summarizing the background information and recent research findings on the virus-alternations of nuclear pore and nucleocytoplasmic transport. The authors have done a good job of listing details for the altered nucleoporions (Nups) by various viruses, which would be helpful for researchers in this field and beyond. However, I hope the authors could address the following major and minor issues before getting it published.

Major issues:

  • In the abstract, the author stated that they will address “the role of viral proteases in the modification of nuclear-cytoplasmic transport in viruses with cytoplasmic replicative cycle and its repercussion in viral replication and in the regulation of the immune response.”. So, as a reviewer and a reader, I would expect that they will organize their manuscript by telling us the following critical sections: 1) the role of viral proteins in the cytoplasmic replicative cycle; 2) the detailed information for viral replication after altering the Nups; and 3) the relationship between altered Nups and the immune response.
  • The section of “Conclusion” is too brief. At least the authors could let us know their perspectives about the near-future and long-term developments of the virus-NPC studies.

Minor issues:

  • In the abstract, it starts by saying “Several viruses…”. It would be better stated as “Various viruses…”.
  • The subtitle “2.2 Bidirectional Transport Nucleus-Cytoplasm” should be “2.2 Bidirectional Nucleus-Cytoplasm Transport”. The error also appeared here and there through the manuscript.
  • “Small molecules of approximately 40-50 kDa” should be “Molecules smaller than approximately 40-60 kDa”.
  • Some typos or misspellings need to be corrected.

Author Response

Reviewer 1

Comments and Suggestions for Authors

In this review manuscript, Jesús-González et al. aim at summarizing the background information and recent research findings on the virus-alternations of nuclear pore and nucleocytoplasmic transport. The authors have done a good job of listing details for the altered nucleoporions (Nups) by various viruses, which would be helpful for researchers in this field and beyond. However, I hope the authors could address the following major and minor issues before getting it published.

Major issues:

  • In the abstract, the author stated that they will address “the role of viral proteases in the modification of nuclear-cytoplasmic transport in viruses with cytoplasmic replicative cycle and its repercussion in viral replication and in the regulation of the immune response.”. So, as a reviewer and a reader, I would expect that they will organize their manuscript by telling us the following critical sections: 1) the role of viral proteins in the cytoplasmic replicative cycle; 2) the detailed information for viral replication after altering the Nups; and 3) the relationship between altered Nups and the immune response.

Repply: Due to the fact that there is few information regarding the relationship between altered NUPS and the immune response. We remove the statement 3 although some examples of this topic are included in the review.

  • The section of “Conclusion” is too brief. At least the authors could let us know their perspectives about the near-future and long-term developments of the virus-NPC studies.

Reply: Thank you for your comment. The conclusion has been extended including perspectives raised by this review (Line 440 to 460).

Minor issues:

  • In the abstract, it starts by saying “Several viruses…”. It would be better stated as “Various viruses…”.

Reply: Thank you for your comment. Your suggestion has been added on the line 12.

  • The subtitle “2.2 Bidirectional Transport Nucleus-Cytoplasm” should be “2.2 Bidirectional Nucleus-Cytoplasm Transport”. The error also appeared here and there through the manuscript.
  • Reply: Thank you for your comment. Your suggestions have been addressed on line 71 and throughout the manuscript. “Small molecules of approximately 40-50 kDa” should be “Molecules smaller than approximately 40-60 kDa”.

Reply: Thank you for your comment. Your suggestions have been added on line 73.

  • Some typos or misspellings need to be corrected.

Reply: Thank you for your comment. We have done an extended grammar and spelling check of the text.

Thank you for your comments and your time in reviewing this paper.

Reviewer 2

Comments and Suggestions for Authors

The review by De Jesús-González et al. provides a comprehensive overview about the alteration of the nuclear core complex by a number of viruses through their protease enzymes to promote viral replication. The article is generally well-presented and the topic highlights the importance of considering the interplay between viral components and host factors in strategies aiming at the development of antiviral agents.

Reply: Thank you for your comment.

1) in lines 126 and 127, the authors mention that the viral proteins C, NS1, and NS5 were identified in the nucleus of Vero cells, but for NS3 and NS4B, the types of cells are not mentioned. Please include the types of cells. Is it expected to have variation in the nuclear distribution of viral components between different cell types?

Reply: Thank you for your comment. Your suggestions have been added on lines 129 to 131. “Other proteins located in the nucleus of human lung carcinoma (A549) infected cells are NS2A, NS3, and NS4A; moreover, the NS3 protein also locates in the nucleus of Huh7 cells during DENV infection.”

Is it expected to have variation in the nuclear distribution of viral components between different cell types?

There is probably a difference in the nuclear distribution of viral proteins in the different cell lines, mainly depending on the intrinsic characteristics of each cell line, such as different proteomic profiles, causing some transport proteins to be more expressed or decreased in some cell lines, which would impact on the transport of viral proteins.

2) in line 41, please correct the sentence. The flaviviral protease does not cleave between basic residues. It cleaves between a basic residue and another amino acid such as serine or threonine.

Reply: Thank you for your comment. Your suggestions have been added on lines 145 and 146 “NS3 protease cleaves between a basic residue (Arg-Lys), and another amino acid such as serine or threonine”.

3) in Figures 3 and 4, it would be helpful to include the indication of the colors used in representing parts of the protein structure by including a color code in the caption.

Reply: Thank you for your comment. A color code was defined for the protein structures.

A few typing errors and punctuation inconsistencies need to be considered by the authors:

  • the term "positive strand" or ""positive-strand" appears differently through the text, an example is in line 34 vs. 44.

Reply: Thank you for your comment. The term “positive-strand” was homogenized in the text, line 35.

  • line 40, please unify the citation, whether after the punctuation at the end of the sentence or before it.

Reply: Thank you for your comment. Citations were homogenized with a punctuation mark after them, line 41.

  • line 43, the word "of" is missing after degradation

Reply: Thank you for your comment. Your suggestion has been added on line 44.

  • some commas should be removed, an example is in line 40 and 52

Reply: Thank you for your comment. Commas have been removed in line 41 and 53 as well as others throughout the manuscript.

  • in Figure 2, the word can now is repeated

Reply: Thank you for your comment. In figure 2 "can now" was removed.

  • in line 144, the word "Flaviviruses" is capitalized and not in other parts of the text as in line 155. Please unify.

Reply: Thank you for your comment. The word "Flaviviruses" was homogenized in the text.

  • the term "2A pro" or "2Apro" appears differently in the text. Please unify.

Reply: Thank you for your comment. The term “2Apro” was homogenized in the text.

Thank you for your comments and your time in reviewing this paper.

Reviewer 2 Report

The review by De Jesús-González et al. provides a comprehensive overview about the alteration of the nuclear core complex by a number of viruses through their protease enzymes to promote viral replication. The article is generally well-presented and the topic highlights the importance of considering the interplay between viral components and host factors in strategies aiming at the development of antiviral agents.

1) in lines 126 and 127, the authors mention that the viral proteins C, NS1, and NS5 were identified in the nucleus of Vero cells, but for NS3 and NS4B, the types of cells are not mentioned. Please include the types of cells. Is it expected to have variation in the nuclear distribution of viral components between different cell types?

2) in line 41, please correct the sentence. The flaviviral protease does not cleave between basic residues. It cleaves between a basic residue and another amino acid such as serine or threonine.

3) in Figures 3 and 4, it would be helpful to include the indication of the colors used in representing parts of the protein structure by including a color code in the caption.

A few typing errors and punctuation inconsistencies need to be considered by the authors:

  • the term "positive strand" or ""positive-strand" appears differently through the text, an example is in line 34 vs. 44.
  • line 40, please unify the citation, whether after the punctuation at the end of the sentence or before it.
  • line 43, the word "of" is missing after degradation
  • some commas should be removed, an example is in line 40 and 52
  • in Figure 2, the word can now is repeated
  • in line 144, the word "Flaviviruses" is capitalized and not in other parts of the text as in line 155. Please unify.
  • the term "2A pro" or "2Apro" appears differently in the text. Please unify.

Author Response

Reviewer 2

Comments and Suggestions for Authors

The review by De Jesús-González et al. provides a comprehensive overview about the alteration of the nuclear core complex by a number of viruses through their protease enzymes to promote viral replication. The article is generally well-presented and the topic highlights the importance of considering the interplay between viral components and host factors in strategies aiming at the development of antiviral agents.

Reply: Thank you for your comment.

1) in lines 126 and 127, the authors mention that the viral proteins C, NS1, and NS5 were identified in the nucleus of Vero cells, but for NS3 and NS4B, the types of cells are not mentioned. Please include the types of cells. Is it expected to have variation in the nuclear distribution of viral components between different cell types?

Reply: Thank you for your comment. Your suggestions have been added on lines 129 to 131. “Other proteins located in the nucleus of human lung carcinoma (A549) infected cells are NS2A, NS3, and NS4A; moreover, the NS3 protein also locates in the nucleus of Huh7 cells during DENV infection.”

Is it expected to have variation in the nuclear distribution of viral components between different cell types?

There is probably a difference in the nuclear distribution of viral proteins in the different cell lines, mainly depending on the intrinsic characteristics of each cell line, such as different proteomic profiles, causing some transport proteins to be more expressed or decreased in some cell lines, which would impact on the transport of viral proteins.

2) in line 41, please correct the sentence. The flaviviral protease does not cleave between basic residues. It cleaves between a basic residue and another amino acid such as serine or threonine.

Reply: Thank you for your comment. Your suggestions have been added on lines 145 and 146 “NS3 protease cleaves between a basic residue (Arg-Lys), and another amino acid such as serine or threonine”.

3) in Figures 3 and 4, it would be helpful to include the indication of the colors used in representing parts of the protein structure by including a color code in the caption.

Reply: Thank you for your comment. A color code was defined for the protein structures.

A few typing errors and punctuation inconsistencies need to be considered by the authors:

  • the term "positive strand" or ""positive-strand" appears differently through the text, an example is in line 34 vs. 44.

Reply: Thank you for your comment. The term “positive-strand” was homogenized in the text, line 35.

  • line 40, please unify the citation, whether after the punctuation at the end of the sentence or before it.

Reply: Thank you for your comment. Citations were homogenized with a punctuation mark after them, line 41.

  • line 43, the word "of" is missing after degradation

Reply: Thank you for your comment. Your suggestion has been added on line 44.

  • some commas should be removed, an example is in line 40 and 52

Reply: Thank you for your comment. Commas have been removed in line 41 and 53 as well as others throughout the manuscript.

  • in Figure 2, the word can now is repeated

Reply: Thank you for your comment. In figure 2 "can now" was removed.

  • in line 144, the word "Flaviviruses" is capitalized and not in other parts of the text as in line 155. Please unify.

Reply: Thank you for your comment. The word "Flaviviruses" was homogenized in the text.

  • the term "2A pro" or "2Apro" appears differently in the text. Please unify.

Reply: Thank you for your comment. The term “2Apro” was homogenized in the text.

Thank you for your comments and your time in reviewing this paper.

Round 2

Reviewer 1 Report

The authors have addressed my concerns and the quality of the manuscript was greatly improved after revision. I would like to support the publication of the manuscript.

Author Response

Thank you very much for your comments to our manuscript. In this new version, the sentence in 394-397 was modified.